# Continuous Electrochemical Reduction of CO_2_ to Formate: Comparative Study of the Influence of the Electrode Configuration with Sn and Bi-Based Electrocatalysts

**DOI:** 10.3390/molecules25194457

**Published:** 2020-09-28

**Authors:** Guillermo Díaz-Sainz, Manuel Alvarez-Guerra, Angel Irabien

**Affiliations:** Chemical and Biomolecular Engineering Department, University of Cantabria, ETSIIT, Avda. Los Castros s/n, 39005 Santander, Spain; manuel.alvarezg@unican.es (M.A.-G.); angel.irabien@unican.es (A.I.)

**Keywords:** CO_2_ valorization, electroreduction, formate, Sn-based materials, bi-based materials, gas diffusion electrodes (GDEs), catalyst coated membrane electrodes (CCMEs), comparative analysis

## Abstract

Climate change has become one of the most important challenges in the 21st century, and the electroreduction of CO_2_ to value-added products has gained increasing importance in recent years. In this context, formic acid or formate are interesting products because they could be used as raw materials in several industries as well as promising fuels in fuel cells. Despite the great number of studies published in the field of the electrocatalytic reduction of CO_2_ to formic acid/formate working with electrocatalysts of different nature and electrode configurations, few of them are focused on the comparison of different electrocatalyst materials and electrode configurations. Therefore, this work aims at presenting a rigorous and comprehensive comparative assessment of different experimental data previously published after many years of research in different working electrode configurations and electrocatalysts in a continuous mode with a single pass of the inputs through the reactor. Thus, the behavior of the CO_2_ electroreduction to formate is compared operating with Sn and Bi-based materials under Gas Diffusion Electrodes (GDEs) and Catalyst Coated Membrane Electrodes (CCMEs) configurations. Considering the same electrocatalyst, the use of CCMEs improves the performance in terms of formate concentration and energy consumption. Nevertheless, higher formate rates can be achieved with GDEs because they allow operation at higher current densities of up to 300 mA·cm^−2^. Bi-based-GDEs outperformed Sn-GDEs in all the figures of merit considered. The comparison also highlights that in CCME configuration, the employ of Bi-based-electrodes enhanced the behavior of the process, increasing the formate concentration by 35% and the Faradaic efficiency by 11%.

## 1. Introduction

Carbon dioxide (CO_2_) emissions to the atmosphere have gradually increased since the middle of the twentieth century from 326 ppm in 1971 to approximately 417 ppm in 2020 [1]. According to the Sustainable Development Goals (SDGs) established by the United Nations and particularly, to SDG Number 13 defined as Climate Action, CO_2_ is the most important human-emitted greenhouse gas in the atmosphere. Therefore, shrinking the CO_2_ emissions should be considered as one of the most important priorities in the current century to mitigate climate change [2]. 

Different strategies can be considered to reduce CO_2_ emissions to the atmosphere, such as improving the energy efficiency or developing renewable energy sources and related technologies [3,4]. In this sense, Carbon Capture, Storage and Utilization (CCSU) [5,6,7,8,9] have been suggested as promising approaches for mitigating climate change. In particular, the possibility of converting captured CO_2_ into fuels and useful industrial chemicals has been especially pointed out [10,11]. 

Indeed, CO_2_ can be transformed to value-added chemical products by several routes, such as thermochemical processes or mineralization [12], biological transformation [13], electrochemical [14] or photochemical/photoelectrochemical conversion [15]. Among these approaches, the electrochemical reduction of CO_2_ toward chemicals with value added has been suggested as an excellent way to store energy from renewable sources in the form of chemical products [16,17]. 

In this respect, different valued-added products [18,19] can be obtained by CO_2_ electrocatalytic reduction such as carbon monoxide (CO) [20,21], formic acid (HCOOH) or formate (HCOO^−^) [22,23] or methanol (CH_3_OH), ethylene (C_2_H_4_), and methane (CH_4_) [24,25,26,27,28]. 

Among these products, HCOOH or HCOO^−^, depending on pH, are used as raw material in several industries (leather tanning, animal feed, steel pickling, or pharmaceutical) [29]. Furthermore, this product has been recommended as an interesting fuel for low-temperature fuel cells [30,31], as well as a promising hydrogen carrier [32,33]. According to the literature [22,34,35], electrocatalysts of different nature, different electrode configurations, and different electrochemical reactors have been used for studying the electrocatalytic reduction of CO_2_ to HCOOH/HCOO^−^. On the one hand, copper (Cu) [36], cobalt (Co) [37], molybdenum (Mo) [38], lead (Pb) [39,40,41], indium (In) [42,43,44], palladium (Pd) [45,46], and especially tin (Sn) [47,48,49,50,51,52,53,54] and bismuth (Bi) [55,56,57,58,59,60,61,62] are the most common catalysts investigated for the selective electrochemical reduction of CO_2_ to HCOOH/HCOO^−^. On the other hand, the electrocatalyst can be used in different electrode configurations, such as in the form of a metal plate, a Gas Diffusion Electrode (GDE), or a Catalyst Coated Membrane Electrode (CCME), operating with different electrochemical reactor configurations and operating conditions. 

Moreover, an important problem in CO_2_ electroreduction is that many different variables may have an influence on the results, which makes it very difficult to compare the results of studies carried out in different experimental conditions. Although there is a great number of studies that focused on the CO_2_ electroreduction to obtain HCOOH/HCOO^−^ using electrocatalysts of diverse nature or cathode configuration, few of them focused on thorough comparisons of the performance of the electrochemical process operating with different catalyst materials and electrode configurations with the same experimental setup and operating conditions [63,64,65]. In this context, this work aims to rigorously compare the behavior of the CO_2_ electroreduction to obtain HCOO^−^ using the same experimental setup but employing electrocatalysts of a different nature, (i) Sn carbon-supported nanoparticles (Sn/C NPs) and (ii) Bi carbon-supported nanoparticles (Bi/C NPs), and under different electrode configurations: (i) GDEs and (ii) CCMEs. The experimental data needed to achieve the aim of manuscript have been previously published after many years of research of our group. A comparative assessment is performed in terms of different relevant figures of merit, including HCOO^−^ concentration, HCOO^−^ rate, Faradaic efficiency for HCOO^−^ (FE), and consumption of energy per kmol of HCOO^−^.

## 2. Comparative Study of the Electrode Configurations: GDE–CCME

This section will compare the performance of GDE and CCME electrode configuration for the electroreduction of CO_2_ to HCOO^−^, employing the same experimental setup and the same electrocatalyst. The comparison will be made first considering the same Sn-based catalyst and then using the same Bi-based catalyst.

### 2.1. Sn-Based Electrodes

The results of different experiments carried out with the same catalyst (Sn NPs) using different working electrode configurations—GDEs (Sn/C-GDEs) and CCMEs (Sn/C-CCMEs)—are reported and discussed in references [66,67], respectively. A summary of the main results of the performance with both configurations is summarized in Table 1. Although the operating conditions were the same, depositing directly the catalytic material over the membrane (Sn/C-CCME) instead of depositing the catalyst over a carbonaceous support (Sn/C-GDE) improved the performance of the electrochemical process, as shown in Table 1. Noteworthy results in terms of HCOO^−^ concentration and energy consumption per kmol of HCOO^−^ using the Sn-based catalyst can be obtained using Sn/C-CCME as a cathode in the electrochemical reactor, with values of 19.2 g·L^−1^ and 244 kWh·kmol^−1^, respectively, which represent noticeably better results than those with the Sn/C-GDE configuration. This improvement can be in part attributed to the better behavior of the electrochemical reactor using a solid polymer electrolyte (with the Sn/C-CCME configuration) instead of a liquid electrolyte (with the Sn/C-GDE configuration). This may be attributed to the fact that the humidified CO_2_ gas input stream enhances the delivery of gaseous CO_2_ to the working electrode surface, and therefore, the solubility limitation of CO_2_ in the electrolyte.

Despite the improvement in the HCOO^−^ concentration and the energy consumption per kmol of HCOO^−^, the HCOO^−^ rate and the FE for HCOO^−^ did not show a substantial enhancement. As can be seen in Table 1, the FE for HCOO^−^ was very similar in both configurations (42.3 and 49.4%), and the HCOO^−^ rate obtained using the Sn/C-CCME configuration decreased from 4.38 to 1.15 mmol·m^−2^·s^−1^ because of the low values of current density supplied to the electrochemical reactor using the Sn/C-CCME cathode (45 mA·cm^−2^) configuration with respect to Sn/C-GDE configuration (90, 150, and 200 mA·cm^−2^). In this sense, the current density employed in Sn/C-CCME was fixed to that value, since an increase in this variable results in higher cell potentials with respect to the use of Sn/C-GDE at the same operating conditions.

Thus, it can be concluded that an Sn/C-GDE configuration is more suitable for operating at higher current densities, and therefore achieving higher rates of CO_2_ reduction and formation of product, but the process results are more energetically favorable under an Sn/C-CCME configuration.

### 2.2. Bi-Based Electrodes

In Table 2, the comparison between the GDE configuration (Bi/C-GDE) [68] and the CCME configuration (Bi/C-CCME) [69], employing the same Bi/C NPs as electrocatalyst, is carried out, in terms of HCOO^−^ concentration, HCOO^−^ rate, FE for HCOO^−^, and the energy consumption per kmol of HCOO^−^. 

It is noteworthy that Bi/C-GDEs were able to operate with a high current density up to 300 mA·cm^−2^, obtaining an HCOO^−^ concentration of 5.2 g·L^−1^ with an FE for HCOO^−^ of 70.6%, an HCOO^−^ rate of 10.97 mmol·m^−2^·s^−1^, and an energy consumption per kmol of HCOO^−^ of 410 kWh·kmol^−1^ of HCOO^−^. Nevertheless, the possibility of obtaining higher HCOO^−^ concentration (18.0 g·L^−1^) with Bi/C-GDEs was by means of lowering the catholyte flow and at the expense of a decrease in both FE for HCOO^−^ (45.1%) and HCOO^−^ rate (4.67 mmol·m^−2^·s^−1^). As illustrated in Table 2, working with a Bi/C-CCME configuration, promising results can be achieved in terms of HCOO^−^ concentration, FE for HCOO^−^, and energy consumption per kmol of HCOO^−^. The best result using a Bi/C-CCME configuration allowed obtaining an HCOO^−^ concentration approximately 44% higher with respect to the highest HCOO^−^ concentration obtained using Bi/C-GDE, and simultaneously, the FE for to HCOO^−^ is approximately 21.5% higher with an important saving in the energy consumption per kmol of HCOO^−^ of around 50%. Similarly, as was found for Sn-based electrodes, the HCOO^−^ rate decreased using a Bi/C-CCMEs configuration with respect to the Bi/C-GDE configuration because of the lower current densities supplied to the electrochemical filter press by potentiostat–galvanostat.

Finally, as can be concluded from the analysis of the results in Table 2, the Bi/C-CCME configuration improves the performance of the electrochemical conversion of CO_2_ to HCOO^−^ in terms of HCOO^−^ concentration, FE for HCOO^−^, and the energy consumption per kmol of HCOO^−^. However, the HCOO^−^ rate did not show an important improvement for the electrocatalytic reduction of CO_2_ to HCOO^−^ because the CCME configuration did not allow operating with high current densities values. 

## 3. Comparative Study of the Catalyst Nature

This section is focused on the comparison of the performance of the CO_2_ electroreduction to obtain HCOO^−^ employing two electrocatalysts of different nature: Sn carbon-supported nanoparticles (Sn/C NPs) and Bi carbon-supported nanoparticles (Bi/C NPs), considering the same experimental setup and the same electrode configuration. The comparison of the two catalysts will be made first using a GDE configuration and then employing a CCME configuration.

### 3.1. Sn vs. Bi in GDE Configuration

A comparative study of the behavior of Sn/C NPs and Bi/C NPs in the form of a GDE is performed. For a clearer visualization, the results obtained using the GDE configurations with Sn/C NPs and Bi/C NPs are summarized in Figure 1 and Figure 2.

As explained in Section 2, the current density and the catholyte flow per geometric area are considered the most influential variables in the performance of the process for the electrochemical conversion of CO_2_ to HCOO^−^ with GDEs The influence of current density in FE for HCOO^−^ (Figure 1a) and the HCOO^−^ rate (Figure 1b) with both electrocatalytic materials can be clearly studied in Figure 1. The current density ranged between 90 and 300 mA·cm^−2^ operating with the same catholyte flow (5.7 mL·min^−1^) to allow a fair comparison. Nevertheless, it was not feasible to supply current densities higher than 200 mA·cm^−2^ using Sn/C-GDEs because of the high cell potential obtained operating with this current density (>4 V). Despite the decrease in the FE for HCOO^−^, from 69 to 55% (with Sn/C-GDEs as working electrode) and from 92 to 80% (with Bi/C-GDEs as working electrode), when the current increased from 90 to 200 mA·cm^−2^, an increase in the HCOO^−^ rate was observed. Moreover, it is important to point out that a noteworthy result in terms of FE for HCOO^−^, of approximately 70%, was achieved using Bi/C-GDEs supplied with a current density of 300 mA·cm^−2^.

As illustrated in Figure 1b, on the one hand, the HCOO^−^ rate using Sn/C-GDEs increased approximately 70% (from 3.23 to 5.61 mmol·m^−2^·s^−1^) when the current density increased from 90 to 200 mA·cm^−2^. In contrast, HCOO^−^ rates up to 8.33 mmol·m^−2^·s^−1^ were obtained using Bi/C-GDEs, which was approximately 50% higher than those with Sn/C-GDE under the same operating conditions and with a current density of 200 mA·cm^−2^. Nevertheless, the highest HCOO^−^ rate was obtained with a current density of 300 mA·cm^−2^ (with Bi/C-GDEs as working electrode), achieving an excellent value of 10.97 mmol·m^−2^·s^−1^.

A summary of the results in terms of HCOO^−^ concentration obtained at different current densities and catholyte flows is represented in Figure 2. On the one hand, operating at a certain same current density (90 or 200 mA·cm^−2^), a higher HCOO^−^ concentration collected in the output stream of the electrochemical filter press cell is obtained if the catholyte flow is lowered from 5.7 to 0.7 mL·min^−1^·cm^−2^. As happened with the FE and the HCOO^−^ rate, the performance of the electrochemical cell in terms of HCOO^−^ concentration is better working with Bi/C-GDEs than with Sn/C-GDEs. The highest HCOO^−^ concentration obtained was 18.0 g·L^−1^, which was 7% higher than operating with the same value of current density and catholyte flow in Sn/C-GDEs (Figure 2). However, working with a current density = 200 mA·cm^−2^, the highest difference between the Bi/C-GDEs and the Sn/C-GDEs was operating with a catholyte flow = 5.7 mL·min^−1^, obtaining an HCOO^−^ concentration of 3.95 g·L^−1^ (using Bi/C-GDE as cathode), which was 50% higher with respect to the value of HCOO^−^ concentration using Sn/C-GDE.

In addition, the comparison of Bi and Sn-based GDEs in terms of energy consumption per kmol of HCOO^−^ is summarized in Table 3. Bi-based electrocatalysts in the form of GDEs resulted in needing less energy consumption per kmol of HCOO^−^ at different current densities and catholyte flows compared with Sn/C GDEs. 

After the analysis of these results, it can be concluded that the use of GDEs with Bi/C NPs gives better results for the electrocatalytic reduction of CO_2_ to HCOO^−^ in terms of all figures of merit studied (FE for HCOO^−^, HCOO^−^ rate, HCOO^−^ concentration and energy consumption per kmol of HCOO^−^). Moreover, it should be highlighted that Bi/C-GDEs allow operating with remarkable values of current density (300 mA·cm^−2^). In the next section, a similar comparison was carried out between Bi/C NPs and Sn/C NPs in the form of CCMEs. 

### 3.2. Sn vs. Bi in CCME Configuration 

As previously discussed in Section 2, the key variables studied in the CCMEs configuration are the temperature and the water flow in the input CO_2_ stream [67,69]. In order to compare the performance of Sn/C-CCMEs and Bi/C-CCMEs, all the experiments considered for the study in this section were carried out operating with a catalyst load = 0.75 mg·cm^−2^ and a current density = 45 mA·cm^−2^. 

First, a comparative of the performance of Sn/C-CCMEs and Bi/C-CCMEs at different temperatures is carried out in terms of Faradaic efficiency for HCOO^−^ (Figure 3a), HCOO^−^ rate (Figure 3b), energy consumption per kmol of HCOO^−^ (Figure 3c), and HCOO^−^ concentration (Figure 3d). The range of temperature studied in both configurations was between 20 and 50 °C. Using either Bi or Sn as electrocatalyst, when the temperature was increased from 20 to 50 °C, all the figures of merit analyzed in this work got worse due to the promotion of the hydrogen evolution reaction. The best results were obtained using Bi-based electrodes operating at ambient condition of temperature, whereas when the temperature was increased to 50 °C, the best performance of the electrochemical process was with the use of Sn/C NPs instead of Bi/C NPs. 

Supplying the CO_2_ stream to the electrochemical reactor with Bi-based CCMEs at a temperature = 20 °C, the FE for HCOO^−^, the HCOO^−^ rate, the energy consumption per kmol of HCOO^−^, and the HCOO^−^ concentration obtained were 47.2%, 1.10 mmol·m^−2^·s^−1^, 312.1 kWh·kmol^−1^, and 22.3 g·L^−1^, respectively. Nevertheless, at the same conditions and using Sn/C-CCMEs, the figures of merit obtained were 47.3%, 1.10 mmol·m^−2^·s^−1^, 226.6 kWh·kmol^−1^, and 18.4 g·L^−1^, respectively. Therefore, although the FE and the HCOO^−^ rate were very similar, the employ of Bi as electrocatalyst gives the best result in terms of HCOO^−^ concentration (21% higher with Bi), and the use of Sn gives better results in term of energy consumption per kmol of HCOO^−^ (27% lower consumption than with Bi). It is interesting to reiterate that raising the temperature from 20 to 50 °C resulted in worse results in terms of all the figures of merit, as illustrated in Figure 3. 

The influence of the water flow in the CO_2_ stream was compared operating at ambient condition of temperature (20 °C) in the CO_2_ input stream to the electrochemical reactor, as summarized graphically in Figure 4. The range of the water flow in the CO_2_ stream studied using both catalyst materials ranged from 0.15 to 1 g·h^−1^. It can be seen in Figure 4a,b that the results of FE and rate obtained using Sn-based electrodes and Bi-based electrodes were very similar in the range of water flow studied. However, the performance of the electrochemical filter press cell was optimized with a water flow in the CO_2_ stream = 0.5 g·h^−1^. On the one hand, using Bi in the form of CCMEs, a FE, a HCOO^−^ rate, an energy consumption, and a HCOO^−^ concentration of 54.8%, 1.28 mmol·m^−2^·s^−1^, 265.8 kWh·kmol^−1^, and 25.9 g·L^−1^, were obtained, respectively. Nevertheless, using Sn/C-CCMEs with the same operating conditions, the HCOO^−^ concentration lowered 35 points in percentage to 19.2 g·L^−1^, keeping similar values in terms of FE, HCOO^−^ rate, and energy consumption (49.4%, 1.15 mmol·m^−2^·s^−1^, and 244 kWh·kmol^−1^, respectively).

In both kinds of electrodes (Sn-based CCMEs and Bi-based CCMEs), raising the water flow in the CO_2_ input stream from 0.5 to 1 g·h^−1^, the behavior of the electrochemical process to give HCOO^−^ in terms of all figures of merit analyzed got worse. On the one hand, the FE and the HCOO^−^ rate (Figure 4a,b) lowered from 54.8% to 47.8% and from 1.28 to 1.11 mmol·m^−2^·s^−1^, respectively, with the use of Bi-based electrodes, and at the same conditions under the Sn-CCME configuration, the FE decreased from 49.4% to 46.3% and the HCOO^−^ rate decreased from 1.15 to 1.08 mmol·m^−2^·s^−1^, respectively. On the other hand, as illustrated in Figure 4c,d, the HCOO^−^ concentration decreased from 25.9 to 22.6 g·L^−1^ and from 19.2 to 17.5 g·L^−1^, whereas the energy consumption per kmol of HCOO^−^ increased to 319.9 and 249 kWh·kmol^−1^ using Bi-based and Sn-based electrodes, respectively.

Furthermore, the results show that the use of Bi-based materials in the form of CCMEs improves the performance of the electrochemical process for the electrocatalytic reduction of CO_2_ to HCOO^−^ in terms of FE for HCOO^−^, HCOO^−^ rate, and HCOO^−^ concentration. However, the employ of Sn/C NPs in the form of CCMEs working at the same operating conditions allows obtaining HCOO^−^ with less energy consumption per kmol of HCOO^−^. 

Finally, further research is still required to overcome current limitations and develop processes with performances that simultaneously optimize all the figures of merit analyzed in this study. In this regard, research efforts should focus on the development of new filter press configurations [70,71] such as the use of a three-compartment electrochemical reactor, bipolar membranes [72], or working electrode configurations [73] to enhance the mass transfer of the reagents and products in the counter and working electrode, and simultaneously address the synthesis of innovative electrocatalysts for both cathode [74] and anode [75] to reduce the energy consumption in both compartments of the electrochemical reactor. 

## 4. Experimental Conditions

All the experiments considered in this comparative analysis were carried out working in a continuous mode with a single pass of the input streams through the electrochemical filter press reactor. The experimental setup included an electrochemical filter press reactor as the core element, a potentiostat–galvanostat, tanks, peristaltic pumps, and a Vapor Delivery Module for operating in a gaseous phase at the cathode side under the CCME configuration. In order to carry out a rigorous comparative study of the electrode configurations, the same electrocatalysts for both configurations were employed: (i) Sn/C-NPs or (ii) Bi/C-NPs. More detail about the synthesis and the characterization of Sn/C NPs and Bi/C-NPs are described in detail in Castillo et al. 2017 [66] and Ávila-Bolívar et al. 2019 [76], respectively.

In contrast, the behavior of the two different electrocatalysts (Sn/C NPs and Bi/C NPs) was compared under the same configuration and experimental setup in both studies: (i) GDEs (Figure 5a), which are described as detailed in Castillo et al. 2017 [66] and Díaz-Sainz et al. 2019 [68], and (ii) CCMEs (Figure 5b), whose fabrication and characterization are described in Díaz-Sainz et al. 2018 [67] and Díaz-Sainz et al. 2020 [69]. In this context, the size of both quasi-spherical Sn/C and Bi/C NPs are 10–15 nm and 9.3 ± 1.6 nm, respectively. The thickness values of the different layers shown in Figure 5 in each of the electrodes compared in this work are summarized in Table 4. In addition, the same operating conditions were employed in the references previously mentioned, as reviewed in Table 5.

In the GDE configuration, the current density and the liquid electrolyte flow per geometric area are considered key variables in the electrochemical conversion of CO_2_ to HCOO^−^, while in the CCMEs configuration, the key variables studied will be the temperature and the water flow in the input CO_2_ stream. It is important to note that in the CCME configuration, due to the characteristics of CCME and unlike in GDEs, it is not feasible to work at high values of current density because this implies huge increases in cell potentials. This is why the comparison in Section 3.2. will be carried out using data at a fixed current density of 45 mA·cm^−2^. In contrast, the temperature is only considered as a key variable for CCMEs because its influence in a GDE configuration is much more limited, but operating in a CCME configuration for the gas-phase electrocatalytic reduction of CO_2_ to HCOO^−^, this variable has an important influence in the amount of water vapor condensed over the CCME surface as well as the water flow in the CO_2_ input stream, which is a crucial aspect in the performance of the process.

Finally, a Dimensionally Stable Anode (DSA) (number 1, Figure 5), a leak-free Ag/AgCl, and a Nafion 117 membrane (number 2, Figure 5) were used as a counter electrode, as a reference electrode, and as a cationic exchange membrane. In addition, the concentration of the HCOO^−^ produced by the electrocatalytic reduction of CO_2_ was analyzed by ion-chromatography technique.

## 5. Conclusions

This work is a comprehensive comparative assessment of different experimental data previously published after many years of research for the electrocatalytic reduction of CO_2_ to HCOO^−^ in different working electrode configurations and electrocatalysts in a continuous mode with a single pass of the inputs through the reactor.

First, the comparison focused on operation with the same electrocatalyst: Sn/C NPs and Bi/C NPs, which were compared in different kinds of working electrode configurations. Considering the same electrocatalyst, the use of CCMEs improves the performance in terms of HCOO^−^ concentration, Faradaic efficiency, and energy consumption when compared with GDEs of that same electrocatalyst. However, the HCOO^−^ rate worsened because of the low values of current densities that had to be supplied to the electrochemical reactor in the CCME configuration. 

Moreover, considering the same operating conditions, a rigorous comparison of both electrocatalysts in the form of GDEs and CCMEs has been carried out. Firstly, using Bi/C-GDEs, the performance of the electrochemical reactor was improved in all the figures of merit analyzed (FE for HCOO^−^, HCOO^−^ rate, energy consumption per kmol of HCOO^−^, and HCOO^−^ concentration) with respect to the Sn/C-GDEs operating at the same conditions. The same comparative study was performed using CCMEs as a cathode configuration. In this new scenario, the use of Bi/C NPs improved the HCOO^−^ concentration in 35%, the FE for HCOO^−^ in 11%, and the HCOO^−^ rate in 11% with respect to the employ of Sn carbon-supported nanoparticles. Nevertheless, the energy consumption per kmol of HCOO^−^ worsened with the use of Bi-based electrodes in 9%.

Finally, despite notable advances achieved, before the electrochemical conversion of CO_2_ to HCOO^−^ could be an industrial reality, further research is still required to optimize all the figures of merit analyzed in this study.

## Figures and Tables

**Figure 1 molecules-25-04457-f001:**
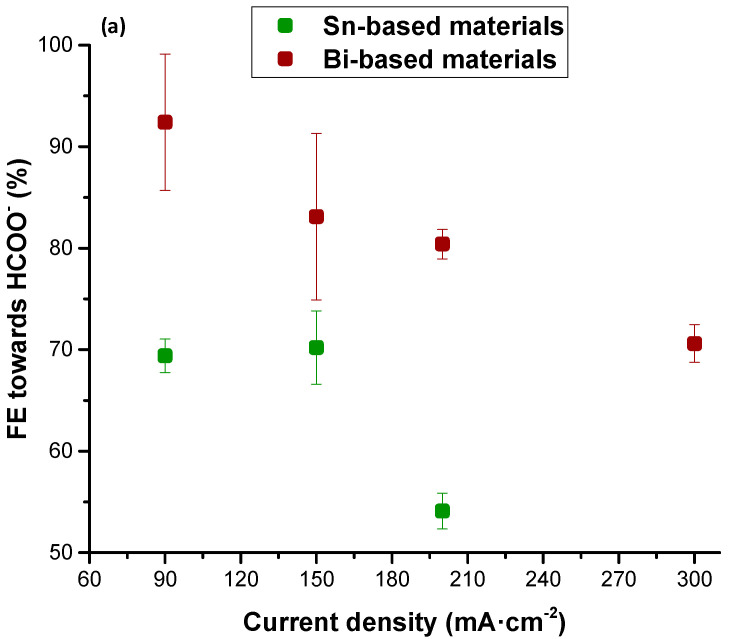
Influence of current density on (**a**) Faradaic Efficiency for HCOO^−^ (%) and (**b**) HCOO^−^ rate (mmol·m^−2^·s^−1^) in the current density range of 90–300 mA·cm^−2^ operating with an electrolyte flow = 5.7 mL·min^−1^. Sn/C-GDEs and Bi/C-GDEs are indicated in green and red, respectively. Data taken from references [66,68]. Sn-C: Sn carbon-supported nanoparticles; Bi/C: Sn carbon-supported nanoparticles.

**Figure 2 molecules-25-04457-f002:**
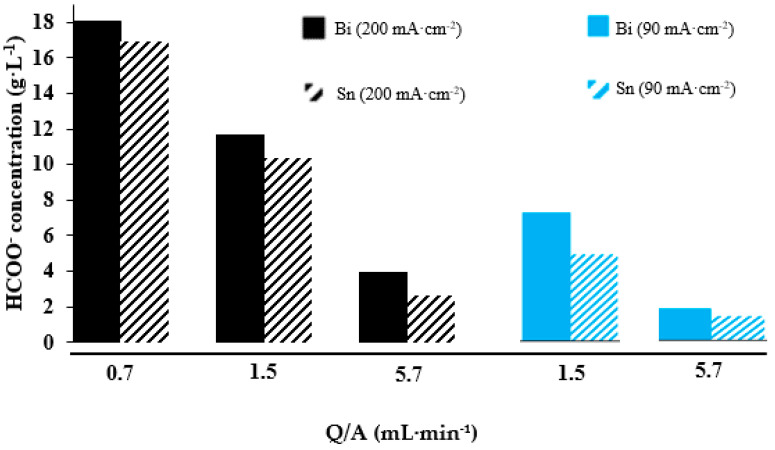
Influence of electrolyte flow on the HCOO^−^ concentration (g·L^−1^) in the electrolyte flow range of 0.7–5.7 mL·min^−1^ applied at different current densities: 90 and 200 mA·cm^−2^ and with different electrocatalyst material (Sn/C NPs and Bi/C NPs). Data taken from references [66,68].

**Figure 3 molecules-25-04457-f003:**
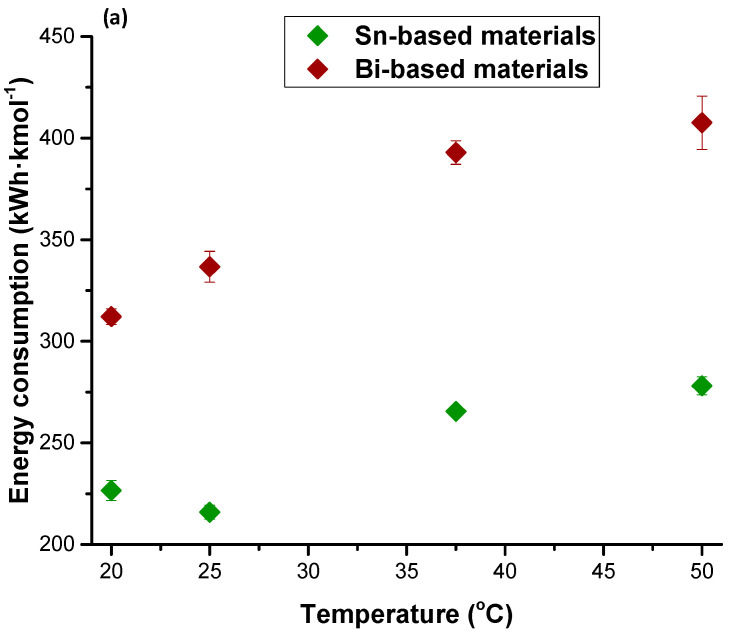
Influence of temperature on (**a**) Faradaic Efficiency for HCOO^−^ (%), (**b**) HCOO^−^ rate (mmol·m^−2^·s^−1^), (**c**) energy consumption per kmol of HCOO^−^ (kWh·kmol^−1^), and (**d**) HCOO^−^ concentration (g·L^−1^) in the temperature range of 20–50 °C applied at a constant current density = 45 mA·cm^−2^, a catalyst loading = 0.75 mg·cm^−2^, and a relative humidity = 100%. Sn/C-CCMEs and Bi/C-CCMEs are indicated in green and red, respectively. Data taken from references [67,69].

**Figure 4 molecules-25-04457-f004:**
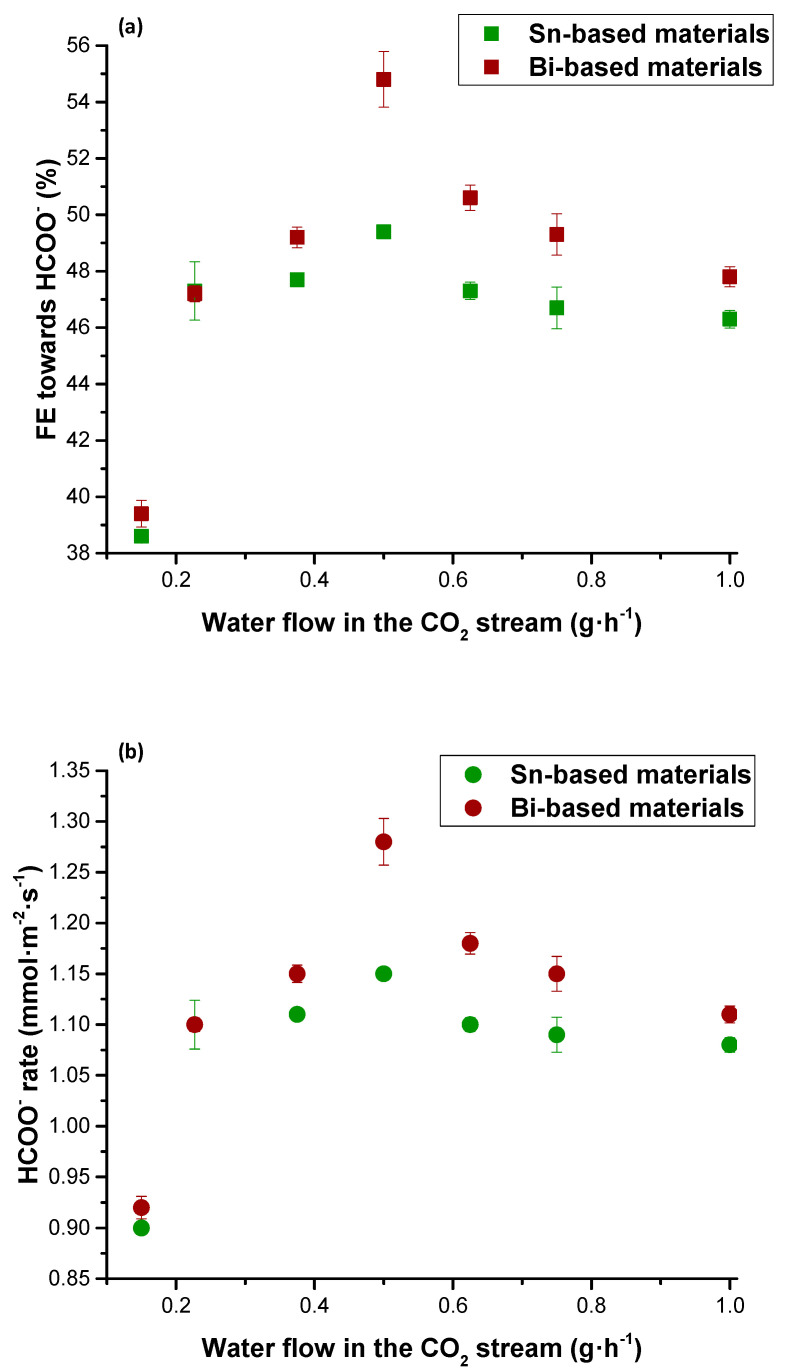
Influence of water flow in the CO_2_ stream on (**a**) Faradaic efficiency for HCOO^−^ (%), (**b**) HCOO^−^ rate (mmol·m^−2^·s−^1^), (**c**) energy consumption per kmol of HCOO^−^ (kWh·kmol^−1^), and (**d**) HCOO^−^ concentration (g·L^−1^) in the water flow range of 0.15–1 g·h^−1^ applied at a constant current density = 45 mA·cm^−2^, a catalyst loading = 0.75 mg·cm^−2^, and a temperature = 20 °C Sn/C-CCMEs and Bi/C-CCMEs are indicated in green and red, respectively. Data taken from references [67,69].

**Figure 5 molecules-25-04457-f005:**
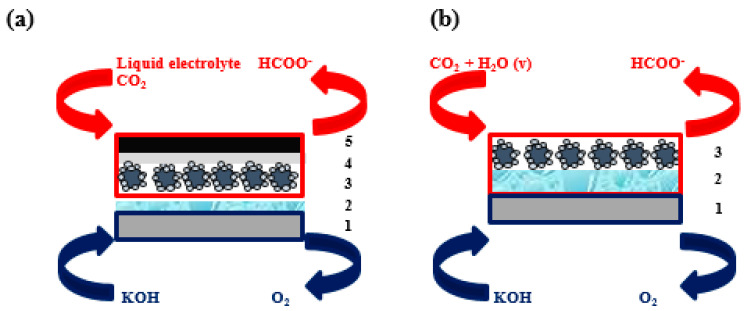
Scheme of (**a**) Gas Diffusion Electrode (GDE) configuration and (**b**) Catalyst Coated Membrane Electrode (CCME) configuration (1: the counter electrode; 2: the cationic exchange membrane; 3: the catalytic layer; 4: the microporous layer, and 5: the carbon support). Please note that there are no elements 4 and 5 in (**b**) because in the CCME configuration, the catalyst is deposited directly on the membrane, avoiding the use of a carbon support and microporous layer.

**Table 1 molecules-25-04457-t001:** Figures of merit (current density, flow in the cathode side, HCOO^−^ concentration, HCOO^−^ rate, Faradaic efficiency (FE) for HCOO^−^ and energy consumption per kmol of HCOO^−^) for the electrocatalytic reduction of CO_2_ to HCOO^−^ employing different Sn-based electrodes (Sn/C-GDEs and Sn/C-CCMEs). Data taken from references [66,67].

Figures of Merit	Configuration				
	Sn/C-GDEs [66]				Sn/C-CCMEs [67]
**Flow in the cathode side**	5.7	5.7	5.7	0.7	~0.008 (0.5 g·h^−^^1^)
**(mL·min^−1^)**					
**Current density**	90	150	200	200	45
**(mA·cm^−2^)**					
**Absolute cell potential**	3.1	3.7	4.0	4.3	2.2
**(V)**					
**HCOO^−^ concentration**	1.5	2.5	2.7	16.9	19.2
**(g·L^−1^)**					
**HCOO^−^ rate**	3.23	5.45	5.61	4.38	1.15
**(mmol·m^−2^·s^−1^)**					
**FE for HCOO^−^**	69.4	70	54.1	42.3	49.4
**(%)**					
**Energy consumption**	239	282	396	513	244
**(kWh·kmol^−1^)**					

**Table 2 molecules-25-04457-t002:** Figures of merit (current density, HCOO^−^ concentration, HCOO^−^ rate, FE for HCOO^−^, and energy consumption per kmol of HCOO^−^) for the electrocatalytic reduction of CO_2_ to HCOO^−^ employing different Bi-based electrodes (GDEs and CCMEs). Data taken from references [68,69].

Figures of Merit	Configuration					
	Bi/C-GDEs [68]					Bi/C-CCMEs [69]
**Flow in the cathode side**	5.7	5.7	5.7	5.7	0.7	~0.008 (0.5 g·h^−1^)
**(mL·min^−1^)**						
**Current density**	90	150	200	300	200	45
**(mA·cm^−2^)**						
**Absolute cell potential**	3.1	3.7	4.2	5.4	4.5	2.7
**(V)**						
**HCOO^−^ concentration**	2.0	3.06	3.9	5.2	18.0	25.9
**(g·L^−1^)**						
**HCOO^−^ rate**	4.31	6.46	8.33	10.97	4.67	1.28
**(mmol·m^−2^·s^−1^)**						
**FE for HCOO^−^**	92.4	83.1	80.4	70.6	45.1	54.8
**(%)**						
**Energy consumption**	177	240	277	410	535	266
**(kWh·kmol^−1^)**						

**Table 3 molecules-25-04457-t003:** Energy consumption per kmol of HCOO^−^ for the electrocatalytic reduction of CO_2_ to HCOO^−^ working with Sn/C-GDEs and Bi/C-GDEs at different operating conditions (current density and catholyte flow). Data taken from references [66,68].

Operating Condition		Energy Consumption (kWh·kmol^−1^ of HCOO^−^)	
Current Density (mA·cm^−2^)	Catholyte Flow (mL·min^−1^)	Sn/C-GDEs [66]	Bi/C-GDEs [68]
90	5.7	239	177
150		282	240
200		396	277
90	1.5	267	186
200		395	364
	0.7	544	535

**Table 4 molecules-25-04457-t004:** Value of thickness of the different layers shown in Figure 5 in each of the electrodes employed in the references [66,67,68,69].

		Sn/C GDEs	Bi/C GDEs	Sn/C CCMEs	Bi/C CCMEs
Thickness	1	2 mm			
	2	183 μm			
	3	50–60 μm	15–20 μm	15 μm	15–20 μm
	4	100 μm	100–125 μm	No microporous layer	
	5	190 μm		No carbon support	

**Table 5 molecules-25-04457-t005:** Value of the different operating conditions taken in the references [66,67,68,69].

Operating Condition	Value
Anolyte flow (mL·min^−1^)	5.7
KOH concentration in anolyte (mol·L^−1^)	1
CO_2_ flow (mL·min^−1^)	200
Catalyst loading (mg·cm^−2^)	0.75
Reaction time (min)	90
Electrode area (cm^2^)	10

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
