# Peer review of "Continuous Electrochemical Reduction of CO2 to Formate: Comparative Study of the Influence of the Electrode Configuration with Sn and Bi-Based Electrocatalysts"

_molecules, 2020, doi:10.3390/molecules25194457_

Round 1

Reviewer 1 Report

The electrochemical conversion of CO2 is a technology that can help address issues of climate change, although the efficiency of the process is still not efficient enough to be widely used on an industrial scale. In the manuscript, the Authors reviewed their contribution to the CO2 conversion struggle. The problem of CO2 conversion is a very broad issue, so I consider it justified to narrow down the review to one technology only.

For the reader's convenience, the authors could add to the work a block diagram of the measuring system used, indicating the elements the performance of which they tried to improve. Such a procedure will enable the reader to find key pieces of information contained in the work more easily. For this purpose, Figure 1 can be extended to provide more information about the system and electrodes applied (device’s and nanoparticles’ dimension, layer thickness, supporting particles dimension, any other important parameters). If the Authors have any photographs or microscopic images of their electrodes they could add them here as well.

Reviewer 2 Report

  1. the same electrocatalysts for both configurations 86 were employed: i) Sn/C-NPs or ii) Bi/C-NPs, do the authors mean amount or? 
  2. Could the author provide any characterization of the catalysts? like sizes, crystallography?

All other aspects are very good.

Reviewer 3 Report

This manuscript studied the CO2 electroreduction to formate by comparing Sn and Bi-based catalysts materials under GDEs and CCMEs and suggested, through a series of the experimental examination, that the use of CCMEs showed much better performance for the same electrocatalyst and that the employ of Bi-based-electrodes presented much enhanced catalytic activities.

But this manuscript did not include more scientifically explanation on why CCME shows better performance than GDE and why Bi-bases catalysts are more reactive than Sn-based catalyst in the CO2 electroreduction to formate.

However, this reviewer believe this manuscript could be highly evaluated in the sense that the research data suggested in this manuscript could be usefully employed in the actual related industries. Moreover, the reviewer judges that the author gave a proper answer to the reviewer's questions, and the revised manuscript is well reorganized by reflecting the answers.

Owing to these two reason, this reviewer recommend the acceptance of this manuscript to the journal of Molecules.

Author Response

This manuscript is a resubmission of an earlier submission. The following is a list of the peer review reports and author responses from that submission.

Round 1

Reviewer 1 Report

The manuscript does not report any new results. It is a review (although very useful) based on data from previous, already published four papers - refs 62, 64-66, masked as 'a comparative study'. The review character should be made more apparent from Abstract and Conclusions. At the same time it would be appropriate to evaluate, for instance, CCME and their application more broadly, considering other published work - for example Hosseini et al. Energy 161 (2018) 1074-1084. This may also involve refs 67-69 - already published work, presented as an outlook in Conclusions. In summary, I recommend the manuscript for publication as Review (see below) but cannot recommend it if the intention was to publish original work.

Reviewer 2 Report

In this paper, the authors compare four set-ups (1. Catalyst: Sn NPs, Device: GDE-based 2. Catalyst: Bi NPs, Device: GDE-based 3. Catalyst: Sn NPs, Device: CCME-based 4. Catalyst: Bi NPs, Device: CCME-based) for CO2RR to formate. For the catalysts part, Sn NPs and Bi NPs share similar sizes. For the device part: GDE and CCME-based device have the same anolyte, CO2 flow, but different configurations and operating conditions. By comparing the catalytic performance (current density, FE, product concentration and so on), the authors conclude that GDE-based device could get higher formate rate and current density, Bi outperforms Sn. However, there are some problems in the paper need to be explained:

  1. All the data in this work come from previous reports (ref. 62 ref. 64 ref. 65 ref. 66) and the authors do not have any new data or updated data. A substantial amount of new information should be provided.

  1. As mentioned above, this paper is highly related to the author's previous work (ref. 64, 65, 66) and could not be considered as follow-up work since there is no new data. Dividing one work into several related manuscripts is not recommended. Please explain it.

  1. GDE and CCME-based device have totally different configurations (the configurations have not described in his paper) and comparing some catalytic data is inappropriate. For instance, the product (formate) remains in anolyte with GDE-device while the product (formate) is separated from the anolyte with CCME-device. Thus, the comparison of formate concentration is meaningless.

  1. The main effort of this paper focus on comparing numbers, not explaining numbers. For instance, a conclusion of part 3.1. (page 4) has been made that Sn-GDE has higher current densities while Sn-CCME is more energy efficient. However, an explanation or speculation is recommended to explain this phenomenon (for example, why CCME is energy efficient).

  1. 'If current density and catholyte flow can be considered key variables working with GDEs, in CCMEs configuration the key variables studied are the temperature and the water flow in the input CO2 stream.' Why the current density in CCMEs and the temperature in GDEs are not important? Please give some explanation.

  1. The FE and energy consumption are highly dependent on applied potential, but the applied potential data have not been presented in this work. The authors claim that high current density could not be achieved with CCME electrode because the applied potential is larger that 4 V (page 5), which indicate the applied potential is different from CCME to GDE (high applied potential sometimes will result in low FE, which could explain why CCME has low FE). Therefore, adding the applied potential data is highly recommended.

  1. Cu (Palmore, ACS Catalysis, 2014), Co (Li, Nature, 2018) and Mo (Han, Angew, 2018)-base materials could also generate high amount of formic acid, which might be added to the introduction.

  1. The alignment of Table 2 is not good and may mislead the reader. The current density in Table 2 is (90, 150, 200 mAcm-2), which is different from Table 3 (90, 200, 300 mAcm-2). According to Figure 1, table 3 has the 150 mAcm-2 data point.

  1. The flow rate in table 2 and table 3 has two different units (should convert gh-1 to mLmin-1). After conversion, 0.5 gh-1 is close to 0.01 mLmin-1, which could explain why CCME has high formate concentration and should be indicated in the manuscript.

  1. the X axis in Figure 2 needs adjustment.

Reviewer 3 Report

In this study, the authors make a comparative assessment in varying figures of merit to examine electrochemical CO2-to-formate performance with Sn and Bi based nanoparticles under gas diffusion electrode (GDE) and catalyst coated membrane electrode (CCME) configurations by reference to their previous works. They demonstrate that the system combined with Bi based nanoparticles and CCME configuration show improved performance in general. However, my main criticism for this works is there really is not much “new” – the data is reproduced, and the discussion of why Bi is more effective and how CCME or GDE designs could be improved is very limited. I believe this work does not meet the typical criterion of novelty to be publishable, unless the authors can provide scientific discussion and insights and/or techno-economic analysis on the comparison.

Detailed Comments:

  1. Page 2, Table 1. I would like to suggest the authors to provide more details in the operating condition such as identity of counter electrode, membrane, reaction time and quantification method in the manuscript to give readers better understanding of the operating system. I feel that the authors too heavily refer their previous works in general.
  2. Page 3, section 3. The authors should prove that each nanoparticle is state-of-the-art catalysts for the fair comparison. It is not clear if the authors want to compare the CO2-to-formate performance based on the intrinsic differences of Sn and Bi nanoparticles in certain electrode configuration. Without providing performance-related physicochemical properties of the catalysts (not just referring to their previous work), it is not clear for the readers to understand if the relatively lower performance of Sn catalysts than Bi results from poor characteristics or lack of thorough nanoparticle design optimization. For example, particle size and shape can be a critical variable to tune the catalytic activities.
  3. Table 2 and 3. The data shown in the tables are just repeated in the following figures from Figure1 to 6 including Table 4.
    1. Page 3, section 3.1. It would be more helpful to examine the energy conversion efficiency of each system with varying experimental conditions given in the Table 2 and 3. Sn/C-GDEs with 0.7 mL min-1 is competitive as compared to Sn/C-CCMEs since both configuration shows similar formate concentration and selectivity. Energy consumption is just a dependent variable as current density increases in the system.
    2. Page 4, section 3.2. Same with above. It will be more comprehensive study if the authors compare energy conversion efficiency of each system with providing cell voltage and current density, correspondingly.
  4. Page 5, section 4.1. It would be helpful for the authors to clarify what other pathways consume current, besides formate formation in Figure 1a. Presumably much of the current goes to hydrogen production, but there’s no mention of catalyst stability either over time, so non-Faradaic currents are possible too.
  5. Page 8, section 4.2. The authors should provide justification why they choose the particular key variables to make comparison on GDEs and CCMEs configuration.
    1. I suggest the author to accumulate Figure 3 and 4 in a single figure (i.e. Figure 3a, 3b, 3c, and 3d for FE, formation rate, energy consumption, and formate concentration, respectively) since the independent variables (temperature) are identical for the current Figure 3 and 4.
    2. The authors should specify the reason behind the loss of formate conversion at higher temperature – presumably this is due to out-competing by hydrogen evolution as has often been reported, but it should be made clear why such large losses are occurring..
    3. I suggest combining Figure 5 and 6 in a single figure with same reason above.
  6. In conclusion section. It tandem with the specification of why losses are occurring where they do, it would be correspondingly helpful for the authors to clarify the reasoning behind their suggestion of further study for new filter press configurations. What losses are due to this assembly and what room is there quantitatively for improvement? What aspects of development in electrode configurations and catalysts do the authors suggest based on the comparison in the manuscript? It would be a stronger contribution if the authors discuss about this point in the main text rather than merely at the end in brief.

Round 2

Reviewer 2 Report

Thanks for the explanations of all my questions. Most of my concerns have been solved (Q3, 4, 5, 7, 8, 9, 10). However, after checking with other reviewers' comments, I find out that lacking novelty (new data) is the main problem of this article. I suggest either change this paper to review format (reviewer 1's suggestion) or add another set of new datapoints (catalyst, electrode configuration or experimental parameter).

Additionally, I still suggest adding 'applied potential' to the manuscript. As I mentioned in Q6, applied potential is highly related to the selectivity (or Faradaic efficiency) of the catalyst. The difference of CO2RR activity between two catalysts may not results from the electrode configuration nor catalyst nature, but the applied potential. I understand that they are fixed current experiments. But the authors could add the time-potential figures to the supporting information.